# Short-Term Caloric Restriction and Subsequent Re-Feeding Compromise Liver Health and Associated Lipid Mediator Signaling in Aged Mice

**DOI:** 10.3390/nu15163660

**Published:** 2023-08-21

**Authors:** Patrick Schädel, Mareike Wichmann-Costaganna, Anna Czapka, Nadja Gebert, Alessandro Ori, Oliver Werz

**Affiliations:** 1Department of Pharmaceutical/Medicinal Chemistry, Institute of Pharmacy, Friedrich Schiller University, D-07743 Jena, Germany; patrick.schaedel@uni-jena.de (P.S.); mareike.wichmann@uni-jena.de (M.W.-C.); anna.czapka@uni-jena.de (A.C.); 2Leibniz Institute for Natural Product Research and Infection Biology, Hans Knöll Institute, D-07745 Jena, Germany; 3Leibniz Institute on Aging—Fritz Lipmann Institute, D-07745 Jena, Germany; nadja.gebert@evotec.com (N.G.); alessandro.ori@leibniz-fli.de (A.O.)

**Keywords:** liver, aging, caloric restriction, inflammation, lipid mediators, PUFA metabolism, lipoxygenase

## Abstract

Aging is characterized by alterations in the inflammatory microenvironment, which is tightly regulated by a complex network of inflammatory mediators. Excessive calorie consumption contributes to age- and lifestyle-associated diseases like obesity, type 2 diabetes, cardiovascular disorders, and cancer, while limited nutrient availability may lead to systemic health-promoting adaptations. Geroprotective effects of short-term caloric restriction (CR) can beneficially regulate innate immune receptors and interferon signaling in the liver of aged mice, but how CR impacts the hepatic release of immunomodulatory mediators like cytokines and lipid mediators (LM) is elusive. Here, we investigated the impact of aging on the inflammatory microenvironment in the liver and its linkage to calorie consumption. The livers of female young and aged C57BL/6JRj mice, as well as of aged mice after caloric restriction (CR) up to 28 days, with and without subsequent re-feeding (2 days), were evaluated. Surprisingly, despite differences in the hepatic proteome of young and old mice, aging did not promote a pro-inflammatory environment in the liver, but it reduced lipoxygenase-mediated formation of LM from polyunsaturated fatty acids without affecting the expression of the involved lipoxygenases and related oxygenases. Moreover, CR failed to ameliorate the secretion of pro-inflammatory cytokines but shifted the LM production to the formation of monohydroxylated LM with inflammation-resolving features. Unexpectedly, re-feeding after CR even further decreased the inflammatory response as LM species were markedly downregulated. Our findings raise the question of how short-term CR is indeed beneficial as a nutritional intervention for healthy elderly subjects and further stress the necessity to address tissue-specific inflammatory states.

## 1. Introduction

The last decades revealed an increase in age- and lifestyle-associated disorders like obesity, type 2 diabetes, cardiovascular diseases, and cancer [1,2,3]. Excessive calorie consumption, which is common in the Western diet, contributes to these disorders [4]. On the other hand, limited nutrient availability without malnutrition, for example, as a result of fasting, induces pro-survival signaling pathways, which are believed to be accompanied by systemic health-promoting adaptations like increased autophagy, resistance to oxidative stress, and alleviation of chronic inflammation and cellular senescence [5,6,7,8]. These geroprotective properties of CR have been observed in several model organisms, ranging from non-vertebrates to rodents to humans [7]. Several studies highlight the role of the liver in sensing micro- and macronutrients and the ability of the hepatic tissue to induce protective mechanisms upon CR, for example, in non-alcoholic fatty liver disease (NAFLD) [9,10]. Specific changes in metabolic and lipidomic signatures alongside adaption of mitochondrial function within the liver lead to decreased oxidative damage and contribute to the observed geroprotective properties of CR [11]. Furthermore, CR in aged rats has been shown to influence the composition of cell types within the liver, for instance, reverting neutrophil frequencies to levels found in early life [12]. Yet, CR has also been linked to stunted growth [13], impaired wound healing [14,15], increased susceptibility to primary intestinal parasite infections [16], and overall reduced survival after infection with viral, bacterial, or parasitic pathogens [17,18]. Furthermore, it has been shown that nutritional memory poses as a limiting factor for the effect CR exerts on the transcriptional and metabolic state when it is started in later life [19]. Consequentially, life- and health-span-extending properties were mostly mitigated, further questioning the benefits of late-life CR [19].

Whilst short-term, late-life CR beneficially impacts the regulation of innate immune receptors and interferon signaling, it does not affect the number of resident innate immune cells [20], and it remains elusive whether this positive effect extends to other immunomodulatory molecules like lipid mediators (LM). In particular, prostaglandins (PG), leukotrienes (LT), and specialized pro-resolving mediators (SPM) that are derived from omega-6 and omega-3 fatty acids, are vastly understudied in the context of CR, while exerting a multitude of functions in maintaining organ homeostasis or driving tissue repair [21,22].

Here, we aim to elucidate whether short-term (4 weeks) CR of 30% with and without subsequent ad libitum access to food (henceforth termed re-feeding, RF) markedly alters the hepatic metabololipidome of 18-month-old C57BL/6JRj mice and rejuvenates the tissue in terms of re-establishing inflammatory biomarker levels commonly found in younger animals. Our data indicate that short-term, late-life CR fails to recover levels of biomarkers for both inflammation and liver health found in younger animals. Furthermore, both CR and re-feeding led to drastically reduced levels of both pro-inflammatory and pro-resolving LM and their precursors, emphasizing that short-term CR in later life acts as an additional stressor on the liver rather than rejuvenating it.

## 2. Materials and Methods

### 2.1. Animals

Female C57BL/6JRj mice were bred and kept at the Fritz Lipmann Institute (FLI) in Jena at a specific pathogen-free animal facility with a 12 h light/dark cycle. Young mice were aged between 2–3 months, and aged mice 18–20 months. The mouse cohort was divided into four experimental groups: (1) young and aged (2) mice with *ad libitum* access to food (ssniff #V1524-786) and water, (3) aged mice that underwent 30% CR (calculated individually as 70% of the animal’s daily food intake) for four weeks, and (4) aged mice that underwent the same caloric restriction but were allowed *ad libitum* access to food and water for 2–3 days after CR, termed re-feeding (RF). One week prior to the caloric intervention, mice undergoing CR were separated into single cages.

### 2.2. Preparation of Liver Lysates

Mice were euthanized using CO_2_. The liver was harvested, washed with PBS, and stored at −20 °C. Lysates were prepared by weighing 40–50 mg liver tissue and adding 20 µL lysis buffer per mg tissue (1% (*v*/*v*) NP-40 (AppliChem, Darmstadt, Germany; A1694), 1 mM sodium orthovanadate (AppliChem; A2196), 10 mM sodium fluoride (AppliChem; A3904), 5 mM sodium pyrophosphate (Sigma Aldrich, St. Louis, MO, USA; S8282), 25 mM β-glycerophosphate (Sigma Aldrich; G9422), 5 mM EDTA (AppliChem; A2937), 25 µM leupeptin (Sigma Aldrich; L2884), 3 mM soybean trypsin inhibitor (Sigma Aldrich; T9128) and 1 mM phenylmethanesulfonyl fluoride (Sigma Aldrich; P7626)). Samples were homogenized using a FastPrep-24™ 5G bead beating homogenizer (M.P. Biomedicals, Irvine, CA, USA) and centrifuged immediately (10 min, 15,000 rpm, 4 °C). Supernatants were transferred into fresh tubes and stored at −20 °C.

### 2.3. Malondialdehyde (MDA) Assay

MDA was quantified in whole liver lysates by using a standard curve within the concentration range of 0.25 to 32 µM of 1,1,3,3-tetramethoxypropane (Sigma Aldrich, 108383). 25 µL of 3 M sodium hydroxide was added to a 100 µL sample or standard and then incubated at 60 °C for 30 min. Then, 0.5 mL of 20% (*w*/*v*) trichloroacetic acid (Sigma Aldrich, T6399) and 1 mL of 0.05 M sulfuric acid were added. Samples were then centrifuged for 10 min at 4 °C and 1000 g. Afterward, 1 mL supernatant was mixed with 0.5 mL of 0.355% (*w*/*v*) thiobarbituric acid (Sigma Aldrich, T5500-25G) and then incubated for 40 min at 90 °C. The incubation was stopped by placing samples on ice. Analysis was performed with 300 µL of samples via HPLC-DAD according to a previously published protocol [23].

### 2.4. Cytokine Quantification

Whole liver lysates were used to determine cytokine and chemokine levels using R&D Systems ELISA Kits for IL-1β (DY401-05), IL-1ra (DY480-05), IL-6 (DY406-05), IL-10 (DY417-05), TNF-α (DY410-05) (R&D Systems, Minneapolis, MN, USA). Assays were performed according to the manufacturer’s instructions, and values were interpolated using individually obtained standard curves.

### 2.5. Proteomics

Whole liver lysates underwent 5 cycles of high-intensity sonication (1 min on, 30 s off, 20 °C) in a Bioruptor system (Diagenode SA, Seraing, Belgium). After centrifugation (4 °C, 10 min, 21,100× *g*) and transfer into fresh tubes, lysis buffer (containing 100 mM N-(2-hydroxyethyl)-piperazine-N′-(2-ethane sulfonic acid) (HEPES, Sigma Aldrich; H3375), 50 mM 1,4-dithiothreitol (DTT, Carl Roth; 6908.3), 2% SDS) was added. Lysates were treated with 10 mM DTT and incubated for 15 min at 45 °C before 15 mM iodoacetamide (Sigma Aldrich; #I1148) was added. After 30 min incubation at room temperature in the dark, the 4-fold volume of ice-cold acetone (Biosolve BV, Valkenswaard, The Netherlands, 010306) was added to the sample. To achieve quantitative protein precipitation, lysates were stored overnight at −20 °C. Samples were centrifuged (4 °C, 30 min, 14,000 rpm), and supernatants were removed. The precipitates were washed twice with 500 µL of ice-cold 80% (*v*/*v*) acetone. Pellets were air-dried before the digestion buffer (1 M guanidine in 100 mM HEPES pH 8.0) was added. Resuspended samples underwent 10 cycles of high-intensity sonication.

Afterward, LysC (Wako Life Sciences, Mountain View, CA, USA) was added at 1:100 (*w*/*w*) enzyme:protein ratio, and digestion proceeded for 4 h at 37 °C under shaking (1000 rpm for 1 h, then 650 rpm). The samples were diluted 1:1 with Milli-Q water and were incubated with a 1:100 (*w*/*w*) amount of trypsin (Promega sequencing grade) overnight at 37 °C, 650 rpm. The digests were then acidified with 10% trifluoroacetic acid and desalted with Waters Oasis^®^ HLB µElution Plate 30 µm (Waters Corporation, Milford, CT, USA) according to manufacturer instructions. Samples were dried down using a speed vacuum centrifuge (Eppendorf Concentrator Plus, Eppendorf, Hamburg, Germany) and redissolved at a concentration of 1 µg/µL in reconstitution buffer (5% (*v*/*v*) acetonitrile, 0.1% (*v*/*v*) formic acid in Milli-Q water).

Digested peptides were analyzed by Data Independent Acquisition (DIA) mass spectrometry. Approximatively 1 μg of reconstituted peptides were loaded on an UltiMate 3000 UPLC system (Thermo Fisher Scientific, Waltham, USA) fitted with a trapping (Waters nanoEase M/Z Symmetry C18, 5 μm, 180 μm × 20 mm) and an analytical column (Waters nanoEase M/Z Peptide C18, 1.7 μm, 75 μm × 250 mm). The outlet of the analytical column was coupled directly to a Q exactive HF (Thermo Fisher Scientific) using the Proxeon nanospray source. Solvent A was water, 0.1% FA and solvent B was 80% (*v*/*v*) acetonitrile, 0.08% FA. Peptides were eluted via a non-linear gradient from 1% to 62.5% B in 131 min. Total runtime was 150 min, including clean-up and column re-equilibration. The S-lens RF value was set to 60. Full scan MS spectra with a mass range 50–1650 *m*/*z* were acquired in profile mode in the Orbitrap with a resolution of 120,000 FWHM. The filling time was set at a maximum of 60 ms with an AGC target of 3 × 10^6^ ions. DIA scans were acquired with 40 mass window segments of differing widths across the MS1 mass range. The default charge state was set to 3+. HCD fragmentation (stepped normalized collision energy; 25.5, 27, 30%) was applied, and MS/MS spectra were acquired with a resolution of 30,000 FWHM with a fixed first mass of 200 *m*/*z* after accumulation of 3 × 10^6^ ions or after filling time of 35 ms (whichever occurred first). Data were acquired in profile mode. For data acquisition and processing, Tune version 2.9 and Xcalibur 4.1 were employed.

Acquired data were processed using Spectronaut Professional v14.9 (Biognosys AG, Zurich, Switzerland). Raw files were searched by directDIA search with Pulsar (Biognosys AG) against the mouse UniProt database (Mus musculus, entry only, release 2016_01) with a list of common contaminants appended, using default settings. Protein group identifications supported by a single peptide were excluded from further analysis. For quantification, default BGS factory settings were used, except Proteotypicity Filter = Only Protein Group Specific; Major Group Quantity = Median peptide quantity; Major Group Top N = OFF; Minor Group Quantity = Median precursor quantity; Minor Group Top N = OFF; Data Filtering = Identified (Q value) and Imputation Strategy = None; Normalization Strategy = Local normalization; Row Selection = Automatic. The candidates and protein report tables were exported from Spectronaut and used for further analysis.

### 2.6. Metabololipidomics via UPLC-MS/MS

Freshly isolated liver (40–50 mg, in PBS (20 µL/mg)) was homogenized using a FastPrep-24™ 5G bead-beating homogenizer (M.P. Biomedicals). After homogenization, lysates were mixed with the same volume of ice-cold methanol. Samples were vortexed, and after centrifugation (10 min, 15,000 rpm, 4 °C), the supernatant was filled up with methanol to a final volume of 3 mL. 10 µL of a deuterium-labeled standard mix (containing 200 nM d8-5-HETE, d4-LTB_4_, d5-LXA_4_, d5-RvD2, d4-PGE_2_ and 10 μM d8-AA) were added to each sample. Solid phase extraction (SPE) was performed by washing with Milli-Q water and n-hexane, and samples were eluted with methyl formate as previously described [24]. Purified samples were evaporated until dryness under continuous N_2_ flow, and the residue was reconstituted in 200 µL MeOH/H_2_O (50:50). After centrifugation (5 min, 15,000 rpm, 4 °C), 80 µL supernatant was used for UPLC-MS/MS analysis. Chromatography was performed using an Acquity TM Ultraperformance LC system (Waters). The LM were eluted using an Acquity UPLC BEH C18 column (Waters, 186002350) at a temperature of 50 °C and a flow rate of 0.3 mL/min. Elution was performed as a gradient elution with a mobile phase consisting of methanol, water, and acetic acid. Over the first 12.5 min, the ratio was increased from 42:58:0.01 (*v*/*v*/*v*) to 86:14:0.01 (*v*/*v*/*v*), then over three min to 98:2:0.01 (*v*/*v*/*v*). Analytes were detected using a QTRAP 5500 mass spectrometer (AB Sciex, Darmstadt, Germany) with a Turbo V source and electrospray ionization. Negative-ionization mode with multiple reaction monitoring (MRM) was used. For quantification, six external deuterated standards were included, and for each analyte, standard curves were obtained.

### 2.7. Data Handling and Statistical Analysis

All experiments were performed as biological replicates on different days, where n represents the number of animals. Results are presented as mean ± standard error of the mean (SEM) unless stated otherwise. Data were analyzed and visualized using GraphPad Prism (Version 9.5.1) or R Studio (version 1.4). Principal component analysis of the metabololipidomics data was performed using the R packages FactoMineR (https://cran.r-project.org/web/packages/FactoMineR/index.html (accessed on 6 April 2023)) and factoextra (https://cran.r-project.org/web/packages/factoextra/index.html (accessed on 6 April 2023)). Outlier testing (ROUT) was performed with a *Q*-value set to 1% and detected outliers were excluded from further analysis. Shapiro–Wilk test (α = 0.05) was conducted for analysis of distribution, and if implied, log-normal distribution was assumed. For the comparison of all groups, unpaired one-way ANOVA was performed. Multiple comparisons were performed using Tukey’s post hoc test (α = 0.05). Brown–Forsythe test was used to determine if standard deviations were significantly different, and if so, a Dunnett’s T3 multiple comparison test was used. The results of all performed significance tests are shown in Appendix A. The criterion for statistical significance is *p* ≤ 0.05, and significance is indicated as: * *p* ≤ 0.05, ** *p* ≤ 0.01, *** *p* ≤ 0.001, **** *p* < 0.0001, n.s. = not significant unless stated otherwise. For volcano plots, proteins with *q*-values ≤ 0.05 and an absolute fold change of log_2_ > 0.58 were considered as significantly affected.

## 3. Results

### 3.1. Aging Has Marginal Impact on the Hepatic Inflammatory Microenvironment

To determine the impact of aging on the inflammatory microenvironment in the liver and later its linkage to calorie consumption, we first assessed the body weight of young and aged female C57BL/6JRj mice across the whole duration (28 days) of the experiment. Interestingly, the body weight of aged mice slightly decreased, whereas young mice slightly gained weight when both were allowed *ad libitum* access to food (Figure 1A).

In order to broadly assess the hepatic microenvironment in both age groups, we prepared whole organ lysates of the liver and employed mass spectrometry-based proteomic profiling to identify comprehensive protein libraries for all experimental cohorts (DDA library: 5165 protein groups, Appendix A). We found an increased variability in the hepatic proteome of aged mice in comparison to young animals (Figure 1B). In detail, aging led to significant changes in the abundance of several proteins related to LM biosynthesis, e.g., several cytochrome P450 (CYP) enzymes (1A2, 2C50, 2C54) which are involved in the conversion of PUFAs (e.g., arachidonic acid) to several monohydroxylated metabolites, were strongly downregulated as a consequence of aging (Figure 1C). LM formation often proceeds in two oxygenation steps of PUFAs, mediated by LOXs and CYPs: the initially formed monohydroxylated precursors serve then as substrates for bioactive di- or trihydroxylated LTs, SPM, and others [25]. Furthermore, typical proteomic markers for macrophage activation (CD14, CD36, CD68 and YM1) were significantly upregulated in aged mice (Figure 1C). Prompted by these results, we screened for a wide selection of markers that are typically expressed by certain innate immune cell subtypes (Figure 1D). Mainly macrophages appear to be present in the liver of both young and aged mice, whereas neither neutrophils nor eosinophils seem to play a major role. F4/80, a common marker for murine macrophages, was completely diminished due to aging (Figure 1D and Appendix A). Interestingly, no significant age-dependent differences in typical Kupffer cell markers CD163 and CD206 were apparent (Figure 1D and Appendix A). To further assess age-driven changes to tissue-resident macrophages (TRM) within the liver, we evaluated proteomic markers typically associated with a pro-inflammatory M1- or M2-like phenotype. Most prominently, YM1 was strongly and significantly increased, whereas other markers were scarcely affected by age (Figure 1D and Appendix A).

**Figure 1 nutrients-15-03660-f001:**
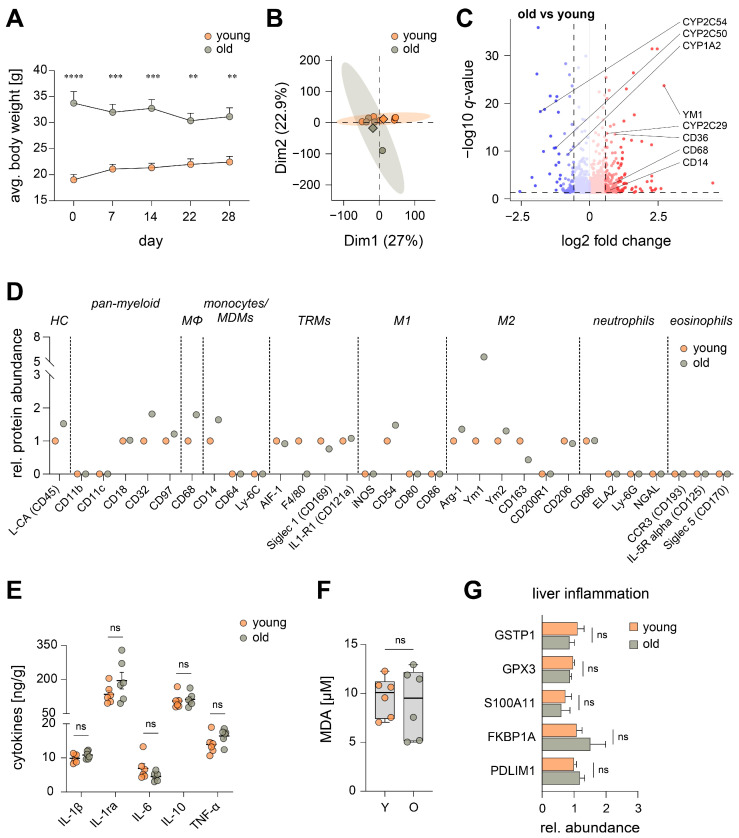
Impact of aging on the hepatic proteome and inflammatory microenvironment of mice with *ad libitum* access to food. (**A**) Average body weight of young (Y) and old (O) mice over the course of the experiment (*n* = 5). (**B**) Principal component analysis of the hepatic proteome. The mean for each experimental cohort is indicated by a rhombus; single data are given as circles. (**C**) Volcano plot of changes to the hepatic proteome of old mice in comparison to young animals (Appendix A, Y: *n* = 6; O: *n* = 4). Dashed lines indicate the cut-off for significance (*q* < 0.05) and absolute fold change (log_2_ > 0.58). (**D**) The relative median abundance of proteomic markers of innate immune cell subtypes in the liver (HC—hematopoietic cells, Mϕ—macrophages, MDM—monocyte-derived macrophages, TRM—tissue-resident macrophages, M1—classically activated (M1) macrophages, M2—alternatively activated (M2) macrophages). Data points at baseline = not detectable (Appendix A, Y: *n* = 5–6; O: *n* = 3–4). (**E**) Cytokine levels in whole liver lysates were measured by ELISA (*n* = 6). (**F**) Hepatic MDA levels in Y and O mice (*n* = 6). (**G**) Relative protein abundance of biomarkers for liver inflammation [26]: glutathione S-transferase P1 (GSTP1), glutathione peroxidase 3 (GPX3), protein S100A11, peptidyl-prolyl cis-trans isomerase (FKBP1A), PDZ and LIM domain protein 1 (PDLIM1) (Appendix A, Y: *n* = 6; O: *n* = 4). Statistics: Data are shown as (**A**,**E**,**G**) mean ± SEM, (**C**,**D**) median or (**F**) median (min to max); *p*- or *q*-values were calculated by one-way ANOVA for multiple comparisons with Tukey’s posthoc test or Brown–Forsythe and Welch ANOVA with Dunnett’s T3 posthoc test (Appendix A) or Spectronaut™ (Appendix A). ** *p* ≤ 0.01, *** *p* ≤ 0.001, **** *p* ≤ 0.0001, ns = not significant.

Surprisingly, these age-related changes in the activation of innate immune cells within the liver were not accompanied by alterations in the release of pro-(IL-1β, IL-6, TNF-α) or anti-inflammatory (IL-1ra, IL-10) cytokines, usually associated with aging [27] (Figure 1E). Next, we sought to correlate our findings to lipid peroxidation levels within the hepatic tissue. Lipid peroxidation occurs as a consequence of increased oxidative damage and is used as a marker for elevated levels of free radicals and oxidative stress, and thus also represents a key biomarker for inflammation [28]. Interestingly, we could not observe a difference in malondialdehyde (MDA) formation in the livers of young and aged mice (Figure 1F). This is in line with biomarkers that are typically screened to assess liver damage related to inflammation, steatosis, and fibrosis [26], which showed no significant differences between both age cohorts (Figure 1G and Appendix A).

### 3.2. Aging Reduces Lipoxygenase-Mediated Oxygenation of Polyunsaturated Fatty Acids

Since the level of several enzymes that convert PUFA into biologically active LM are altered in the liver of aged mice (Figure 1C), we opted to assess the hepatic metabololipidome. Through principal component analysis (PCA), we found no striking differences in the overall LM signature profiles between both age groups (Figure 2A). However, when clustering the LM profile into metabolomes of the respective PUFAs, the average total amount of LM derived from docosahexaenoic acid (DHA) dropped by about 40% as a consequence of aging (Figure 2B), yet no statistical significance was reached due to high variability. Dihomo-γ-linolenic acid (DGLA)- and eicosapentaenoic acid (EPA)-derived LM were slightly reduced (~17%) or increased (~19%), respectively, while metabolites of AA and α-linolenic acid (ALA) were not impacted by age (Figure 2B). In order to obtain detailed insight into the inflammatory status of the liver of both age cohorts, we compared the metabololipidome in the liver of young and aged mice. We found a substantial, yet again not significant decrease of monohydroxylated LM precursors (3.12 to 1.84 ng/g) due to aging, which was independent of the corresponding lipoxygenase (LOX) pathways (Figure 2C), did not translate into bioactive LM (Figure 2C), and had no effect on the ratio of pro-inflammatory versus pro-resolving LM (Figure 2D). In particular, PGE_2_, LTB_4_ isomers and resolvin (Rv)D5, which are commonly associated with inflammaging [29], were not markedly affected by age (Figure 2E). The aforementioned reduction in the absolute amount of monohydroxylated LM precursors (Figure 2A,C) may be due to alterations in the expression of several CYP enzymes due to aging (Figure 1C and Appendix A). Proteomic profiling and Western blot analysis of the biosynthetic enzymes corresponding to pro-inflammatory PG (COX-1/2, mPGES-2) and LT (5-LOX) or to inflammation-resolving SPM formation (15-LOX-1, 5-LOX) revealed no significant age-related changes in their expression levels (Figure 2F,G). This further translates into the composition of the metabololipidome of both age cohorts, which is basically identical with a minor decrease of 12- and 15-LOX products in the liver of aged mice (Figure 2H and Appendix A).

### 3.3. Caloric Restriction Elicits Alterations in the Hepatic Inflammatory Microenvironment of Old Mice

CR is a well-described anti-aging intervention that has been linked to increased lifespan and reduced basal inflammation [30,31]. We sought to identify the consequences of short-term 30% CR for liver health and the hepatic inflammatory microenvironment in aged mice. Aged animals which underwent the CR intervention continuously lost weight during the experiment, while aged mice with *ad libitum* food access (O) essentially maintained their weight (Figure 3A). Of note, aged mice that initially lost weight due to CR quickly regained it given *ad libitum* access to food during a 2-day re-feeding period, as has been previously reported [32]. Interestingly, the hepatic proteome of aged mice who underwent both CR and subsequent RF (designated as CR+RF) strongly overlapped with that of young mice, whereas CR alone did not yield this effect (Figure 3B).

CR ameliorated pro-inflammatory actions of interferon-γ (IIGP1, IFITM3) while potentially increasing the cleavage of PUFA from phospholipids through suppression of inhibitory factors like ANXA3 (Figure 3C). Furthermore, CR altered the metabolism of PUFA in aged mice both through down- (CYP3A16, CYP3A41) and upregulation (CYP4A14, GSTP1, CYP3A25, CYP2D9, CYP2C54) of associated enzymes (Figure 3C). A short period (2 days) of RF (CR+RF group) also up-(CYP2C54, GSTP1, CYP2D9) or downregulated (GSTA3, CYP3A16, CYP2C38, GSTA2) enzymes associated with PUFA metabolism (Figure 3C). Yet, when comparing both experimental groups (CR and CR+RF), we found a substantial reduction upon RF in several CYP450 isoenzymes and other PUFA-metabolizing enzymes (AKR1B7, AKR1B1) along with an increased expression of interferon-associated proteins (IRF5, IIGP1) (Appendix A). Additionally, both CR and CR+RF influenced the level of proteins that scavenge and regulate oxidative stress (SOD1, GPX3, and SRXN1; Figure 3C).

To further assess liver health, we evaluated the influence of CR and RF on typical hepatic damage markers and found that CR-induced patterns were commonly associated with liver inflammation and fibrosis, whereas steatosis markers were not affected (Figure 3D and Appendix A). Subsequent re-feeding alleviated some of the changes in the inflammation pattern but had no effect on fibrosis and steatosis (Figure 3D and Appendix A). Both interventions failed to reestablish youthful levels in most screened liver health markers (Figure 3D and Appendix A).

Screening of proteomic innate immune cell markers revealed changes in macrophage activation by decreased IL1-R1 expression alongside increased Arg-1 and Ym1 levels (Figure 3E and Appendix A). Further, CR reduced typical Kupffer cell markers CD163 and CD206 (Figure 3E and Appendix A). In combination with low levels of F4/80 (Appendix A), this may implicate dwindling Kupffer cell numbers within the tissue. Re-feeding slightly elevated F4/80 and CD163 levels, but overall could not recover youthful levels similarly to CR (Appendix A).

Next, we screened for cytokines typically associated with inflammaging and often used to evaluate the efficacy of CR interventions [33,34]. The levels of pro-inflammatory IL-1β and IL-6 did not change in response to CR, but anti-inflammatory IL-1ra was significantly decreased (Figure 3F). Surprisingly, CR drastically enhanced the release of pro-inflammatory TNF-α, which was ameliorated by RF (Figure 3F). Overall, cytokine levels in the liver of CR+RF mice mostly overlapped with those found in young animals (IL-1β, IL-6), and RF ameliorated CR-induced changes (IL-1ra, TNF-α). Of interest, CR drastically reduced lipid peroxidation, whereas RF reversed MDA levels back to levels found in untreated old mice (Figure 3G).

### 3.4. CR and Subsequent Re-Feeding Diminish PUFA Metabolism in Aged Mice

Since CR and subsequent RF markedly influenced the level of enzymes involved in PUFA metabolism, we assessed the hepatic metabololipidome by UPLC-MS/MS with respect to CR and RF. PCA revealed widely overlapping clusters for all experimental groups, with a higher variability among old mice with ad libitum access to food and rather narrow clusters for both the CR and the CR+RF cohort (Figure 4A). CR reduced the amount of DGLA-derived LM (23%) while raising the levels of DHA-derived LM (23%). The AA, ALA and EPA metabolomes were only marginally decreased as a consequence of CR (Figure 4B, Appendix A). Interestingly, we observed a decrease across all LM for the CR+RF cohort in comparison to ad libitum-fed old mice and even CR mice with the highest discrepancy from youthful levels among all experimental groups (Figure 4B, Appendix A). Yet, we did not observe any changes in the ratio of pro-inflammatory and pro-resolving LM between all experimental groups (Figure 4C). Interestingly, CR slightly increases the amount of monohydroxylated LM precursors due to 5, 12- and 15-lipoxygenation while reducing the overall amount of PG, TX and SPM (Figure 4D).

The most striking differences are caused by RF, which drastically limits the formation of LT, PG, TX and SPM in the liver of old mice (Figure 4D). Additionally, RF vastly diminishes the overall metabololipidome in terms of the formation of both monohydroxylated LM precursors, as well as bioactive LM in comparison to youthful levels (Figure 4D). Furthermore, RF affects the composition of the hepatic metabololipidome by strongly increasing the share of COX and 12/15-LOX products, whereas CR mainly restores 12/15-LOX products to levels of young mice (Figure 4E and Figure 2H). Comparing selected, prominent monohydroxylated 5-, 12- and 15-LOX products from AA, DHA and EPA showed a slight upregulation by CR (except 12-HEPE) and a drastic reduction by RF (except 5- and 15-HEPE) (Figure 4G). Simultaneously, CR reduced the formation of 6-keto-PGF1α and SPM (mainly LXB_5_), whereas RF completely diminished the formation of prominent PG (PGD_2_, PGE_2_, PGJ_2_, 6-keto-PGF_1_α), LTB_4_ and SPM (LXB_5_, MaR2, PDX, RvD5) in the liver of old mice (Figure 4G). Overall, we found that aging had a rather slim effect on the hepatic metabololipidome, while CR shifts PUFA metabolism towards monohydroxylated LM precursors, and subsequent RF diminishes the formation of essentially all LM species (Figure 4F).

We finally sought to correlate our findings on the hepatic metabololipidome to the levels of LM biosynthetic enzymes, but we could not find any significant changes, except for a significant reduction in prostaglandin I synthase (PGIS) in CR (Appendix A), which does not translate into increased product formation (the metabolite of unstable PGI_2_, namely 6-keto-PGF_1_α; Figure 4G). 15-LOX-1 showed a slight upregulation due to CR, which fits well with the increased formation of related monohydroxylated LM (12-HETE, 14-HDHA, 15-HETE, 15, HEPE, 17-HDHA; Figure 4G).

## 4. Discussion

In general, aging is associated with an increase in low-grade, sterile, chronic inflammation, a process termed inflammaging that is exacerbated by high caloric intake, which is common in Western diets [35]. This interrelation stimulated extensive research into CR as a measure to counteract aging and to improve overall health in later stages of life [31,36,37]. While lifespan extension as a consequence of CR has been proven across species, recent findings have shed light on major differences in the efficacy and benefits of CR depending on the age of the individual, the duration of the CR period, and its alignment with the circadian rhythm [19,20,38]. Here, we investigate the effect of short-term CR on the hepatic inflammatory microenvironment in later life and its rejuvenating potential on liver health in mice.

We first assessed the degree of age-related alterations to the hepatic inflammatory microenvironment in female mice at 18 months (old) versus ones at 3 months (young). Proteomic analysis revealed several minor changes to the composition of innate immune cell populations, which hardly translated into noticeable changes in the release of inflammation-related cytokines or levels of lipid peroxidation, as we described in a previous study using male mice [39]. This might be explained by the age of the animals (18 months, in this study, vs. 24 months, in a previous study) and the sex of the animals, as aging has been shown to differ between male and female individuals [40,41]. On the metabololipidomic level, aging mainly caused depletion of monohydroxylated LM precursors that are formed through 5-, 12- and 15-lipoxygenation, while levels of bioactive LM (PG, LT, TX, SPM), as well as the ratio of pro-inflammatory eicosanoids to pro-resolving ones, remained unaffected. In agreement with previous findings, we observed only minor changes in the composition of the metabololipidome due to aging while retaining organ-specific patterns [39]. Overall, we found marginal changes to the inflammatory microenvironment in the liver of 18-month-old female mice, when compared to young, which hardly resembled a typical inflammaging phenotype.

Aside from the inflammatory microenvironment in the liver, aging increased the weight of the animals and altered their overall hepatic proteome, but only minorly affected biomarkers that are related to liver inflammation, steatosis or fibrosis [26]. These latter findings contradict previous studies that describe the attritious effect of aging on liver health [42,43]. Based on our findings, we questioned whether short-term CR at a later age with and without subsequent RF may actually be beneficial for the liver health of old animals or rather acts as an additional stressor, as has been implied in other studies [19,44,45,46].

As expected, short-term CR markedly reduced the body weight of the animals and, in line with previous studies, led to a plethora of changes to the hepatic microenvironment [20,46,47,48]. Overall, short-term CR failed to revert the release of inflammation-related cytokines and LM to levels found in young animals. It rather acted as an additional stressor that exacerbated existing age-associated changes or even caused aberrations that aging itself did not. Consequently, CR led to an elevation of proteomic markers associated with liver inflammation (e.g., GSTP1, GPX3) or liver fibrosis (e.g., TGFBI, TNC) [26], and therefore, rather compromised liver health instead of improving it. Some of these marker patterns were reversed as soon as animals were given *ad libitum* access to food during a 2-day re-feeding phase, implying a rather transient impact of short-term CR. This is in line with a previous assessment that questioned the efficacy of short-term CR in older animals due to the nutritional memory [19]. Similarly, we observed changes in cytokine secretion and lipid peroxidation upon CR, which again were mostly of a transient nature upon re-feeding and reverted back to initial levels found in old animals.

Yet, on the metabololipidomic level, we found that the CR and a short re-feeding period vastly abolished PUFA metabolism for monohydroxylated LM precursors and bioactive PG, LT, and SPM. Some of the observed changes may be correlated to the expression levels of CYP enzymes, as they are known to be affected by dietary interventions [49], and CYP isoenzyme levels were significantly affected by CR and subsequent re-feeding. Since LM, especially SPM, are a vital component of the inflammatory microenvironment and necessary to maintain homeostatic conditions within the tissue [21,22], one may expect a decline in resilience, which may, in turn, be causative for a higher susceptibility towards infection after CR, as has been observed previously [16,17,18]. Particularly, the absence of monohydroxylated LM precursors prevents further production of highly bioactive PG, LT and SPM that impact many regulatory processes in the liver. For example, homeostatic levels of PGE_2_ and other COX-derived prostanoids are necessary for liver health. It has been shown that these hepatoprotective effects include liver regeneration upon damage [50,51] and increased resilience against liver injury [52]. Additionally, SPM that are biosynthesized mainly from EPA and DHA act as immunoresolvents to terminate the inflammatory process, promote the return to homeostasis and have been implicated as beneficial supplements in the treatment of NAFLD [25,53,54].

Taken together, it remains questionable how reliably short-term CR improves the overall liver fitness and resilience in organisms with a regular diet, particularly in the context of inflammation. As of now, it seems that the most striking benefits of CR on liver health are evident in studies employing animals that had been subjected to a high-fat diet prior to CR [55,56] or in clinical studies with obese human participants [57,58]. Furthermore, it has been shown that a macronutritionally balanced diet offers superior health and longevity benefits than caloric restriction [59]. Overall, we did not find convincing evidence that short-term CR in older animals improves liver health, in particular on the metabololipidomic level; it rather acts detrimentally.

## Figures and Tables

**Figure 2 nutrients-15-03660-f002:**
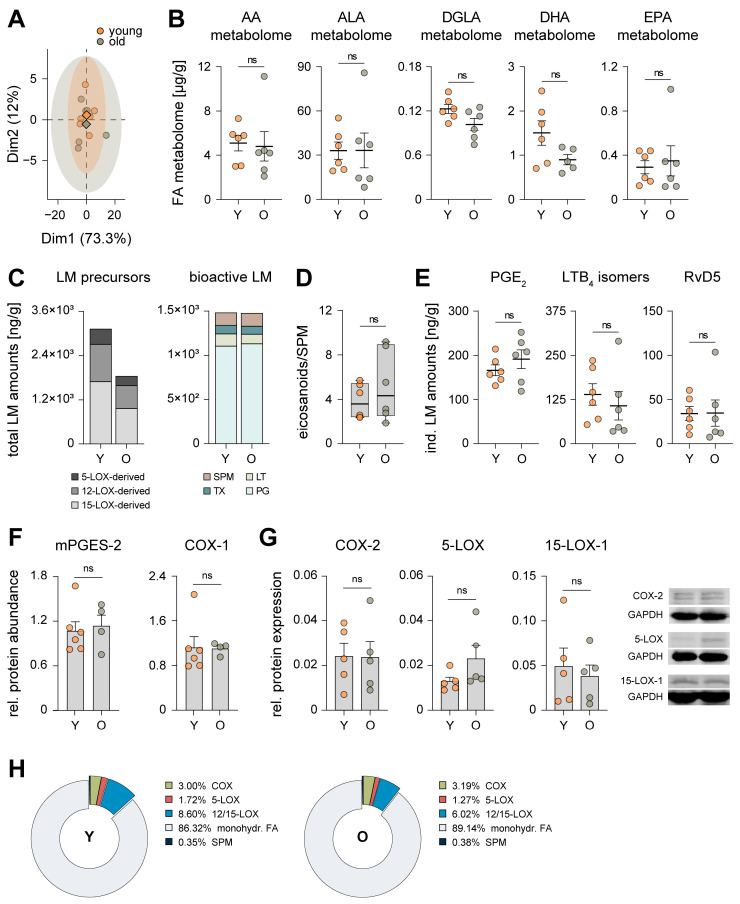
Aging does not cause a pro-inflammatory shift in the hepatic metabololipidome. (**A**) Principal component analysis of the hepatic metabololipidome of young (Y) and old (O) mice (Appendix A, *n* = 6). The mean for each experimental cohort is indicated as rhombus; single data are given as circles. (**B**) Absolute levels of hepatic metabolomes from arachidonic acid (AA), α-linolenic acid (ALA), dihomo-γ-linolenic acid (DGLA), docosahexaenoic (DHA), and eicosapentaenoic acid (EPA) of Y and O mice (Appendix A, *n* = 6). (**C**) Stacked histograms of 5-LOX-, 12-LOX- and 15-LOX-derived monohydroxylated LM precursors and of bioactive prostaglandins (PG), leukotrienes (LT), thromboxane (TX), and specialized pro-resolving mediators (SPM) in the liver of Y and O mice (Appendix A, *n* = 6). (**D**) The ratio between pro-inflammatory eicosanoids (PGE_2_, PGD_2_, LTB_4_, epi-trans-LTB_4_, trans-LTB_4_) and SPM (LXB_5_, PDX, MaR2, RvD5) in the liver of Y and O mice (Appendix A, *n* = 6). (**E**) Levels of PGE_2_, LTB_4_ and its isomers epitrans-LTB_4_ and trans-LTB_4_, and RvD5 in the liver of Y and O mice (Appendix A, *n* = 6). (**F**) Protein levels of mPGES-2 and COX-1 in the hepatic proteome of Y and O mice were determined by proteomic analysis (Appendix A, Y: *n* = 6; O: *n* = 4). (**G**) Expression of COX-2, 5-LOX and 15-LOX-1 relative to GAPDH (housekeeping protein) in the liver of Y and O mice was determined by SDS-PAGE and Western Blot (*n* = 5). (**H**) Composition of the hepatic metabololipidome of Y and O mice clustered according to corresponding LM biosynthetic enzymes (except SPM and monohydroxylated LM). Values are presented as percentages of the overall amount of LM (*n* = 5–6). Statistics: Data are shown as (**B**,**E**–**G**) mean ± SEM, (**C**,**H**) mean and (**D**) median (min to max); *p*-values were calculated by one-way ANOVA for multiple comparisons with Tukey’s posthoc test or Brown–Forsythe and Welch ANOVA with Dunnett’s T3 posthoc test (Appendix A). ns = not significant.

**Figure 3 nutrients-15-03660-f003:**
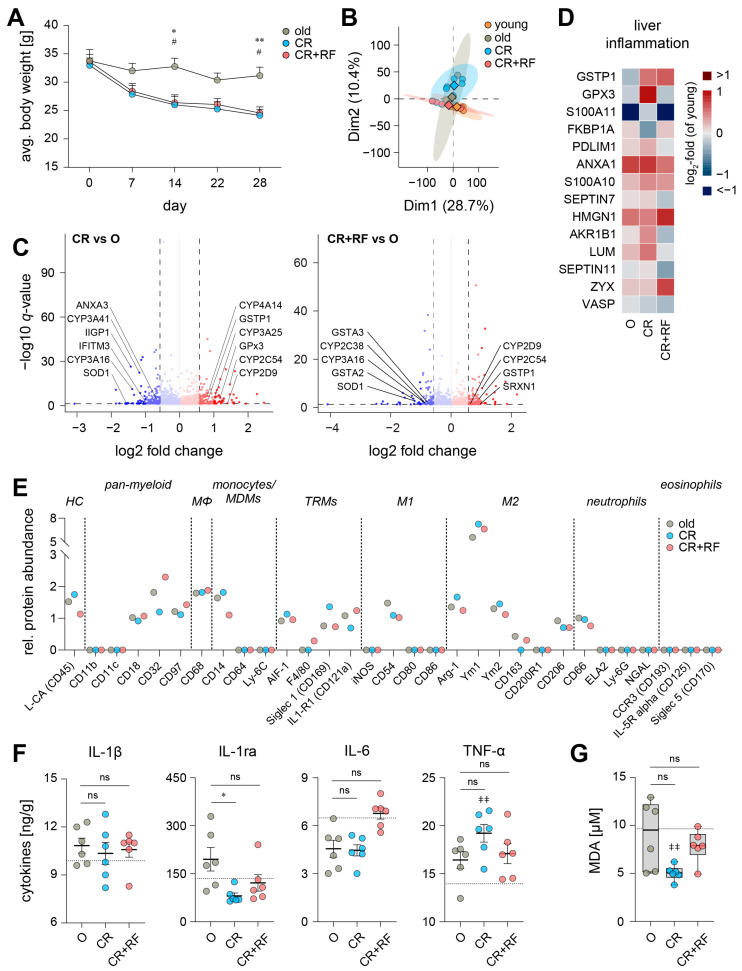
Impact of CR and RF intervention on the hepatic proteome and inflammatory microenvironment. (**A**) Average body weight of old (O), caloric restricted (CR), and re-fed (CR+RF) mice over the course of the experiment. Values are presented as mean ± SEM (*n* = 5). (**B**) Principal component analysis of the hepatic proteome of young (Y), O, CR, CR+RF mice. The mean for each experimental group is indicated by rhombus; single values are given as circles. Volcano plots of changes to the hepatic proteome (**C**) in CR mice and in CR+RF mice compared to old mice (Appendix A, O: *n* = 4; CR, CR+RF: *n* = 6). Dashed lines indicate the cut-off for significance (*q* < 0.05) and absolute fold change (log_2_ > 0.58). (**D**) Relative protein abundance of biomarkers for liver inflammation [26]. Heatmap represents the log_2_-fold change compared to the median in young mice (Appendix A, O: *n* = 4; CR, CR+RF: *n* = 6). (**E**) Relative abundance of proteomic markers of innate immune cell subtypes in the liver (HC—hematopoietic cells, Mϕ—macrophages, MDM—monocyte-derived macrophages, TRM—tissue-resident macrophages, M1—classically-activated (M1) macrophages, M2—alternatively activated (M2) macrophages). Data points at baseline were not detectable (Appendix A, O: *n* = 3–4; CR, CR+RF: *n* = 5–6). (**F**) Cytokine levels in whole liver lysates were assessed by ELISA (*n* = 6). The dotted line represents the mean level in young mice (Figure 1E). (**G**) Hepatic MDA levels in O, CR, CR+RF mice (*n* = 6). The dotted line represents the mean level in young mice (Figure 1F). Statistics: Data are shown as (**A**,**D**,**F**) mean ± SEM, (**C**,**E**) median or (**G**) median (min to max); *p*- and *q*-values were calculated by one-way ANOVA for multiple comparisons with Tukey’s posthoc test or Brown–Forsythe and Welch ANOVA with Dunnett’s T3 posthoc test (Appendix A) or Spectronaut™ (Appendix A). Statistical significance is indicated by asterisks for comparisons between CR and O, by hashes for comparisons between CR+RF and O and by ǂ for comparisons with the levels of young mice. */# *p* ≤ 0.05, **/ǂǂ *p* ≤ 0.01, ns = not significant.

**Figure 4 nutrients-15-03660-f004:**
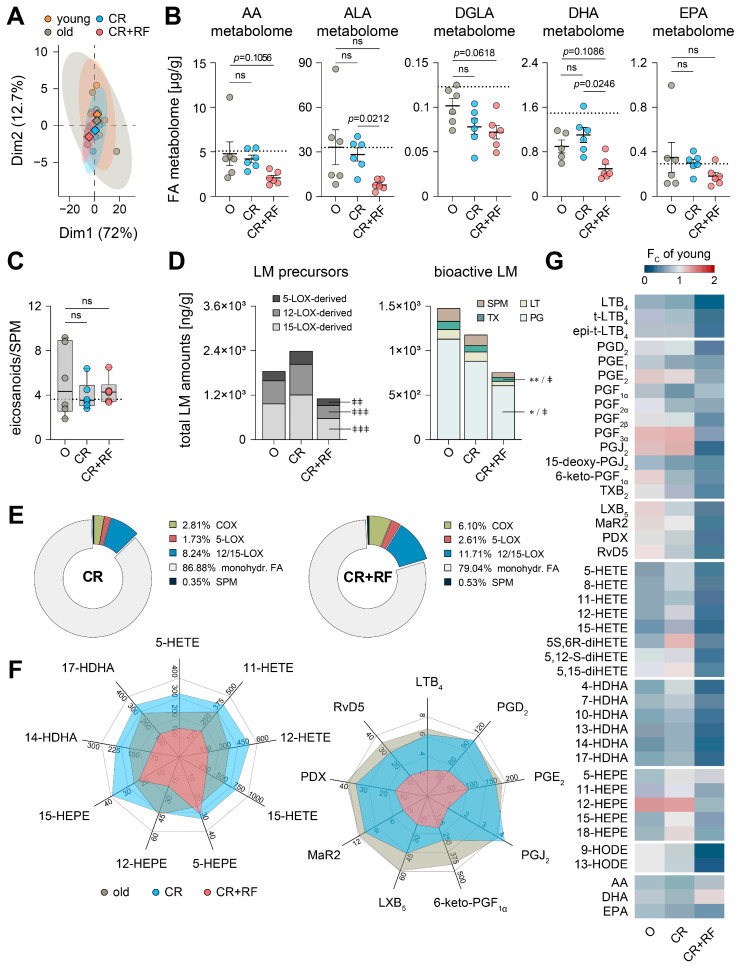
Influence of caloric restriction and subsequent re-feeding on the hepatic metabololipidome of old mice. (**A**) Principal component analysis of hepatic metabololipidome in young (Y), old (O), caloric restricted (CR), and re-fed (CR+RF) mice. The mean for each experimental cohort is indicated as rhombus; single values are given as circles. (**B**) Absolute levels of hepatic metabolomes from arachidonic acid (AA), α-linolenic acid (ALA), dihomo-γ-linolenic acid (DGLA), docosahexaenoic (DHA), and eicosapentaenoic acid (EPA) of O, CR, and CR+RF mice (Appendix A, *n* = 5–6). The dotted line represents the mean level in young (Y) mice (Figure 2B). (**C**) Ratio between pro-inflammatory eicosanoids (PGE_2_, PGD_2_, LTB_4_, epi-trans-LTB_4_, trans-LTB_4_) and SPM (LXB_5_, PDX, MaR2, RvD5) in the liver of O, CR, and CR+RF mice (*n* = 6). The dotted line represents the mean level in young (Y) mice (Figure 2D). (**D**) Stacked histograms of 5-LOX-, 12-LOX- and 15-LOX-derived monohydroxylated LM precursors and of bioactive prostaglandins (PG), leukotrienes (LT), thromboxane (TX), and specialized pro-resolving mediators (SPM) in the liver of O, CR, and CR+RF mice (Appendix A, *n* = 5–6). (**E**) Composition of the hepatic metabololipidome of CR and CR+RF mice clustered according to corresponding LM biosynthetic enzymes (except monohydroxylated PUFA). Values are presented as the percentage of the overall amount of LM (*n* = 5–6). (**F**) Radar charts of the levels of monohydroxylated LM precursors (left panel) and bioactive LM (right panel) in the liver of O, CR, and CR+RF mice (Appendix A, values given as ng/g, *n* = 5–6). (**G**) Heatmap showing changes in LM levels in the liver of O, CR and CR+RF mice in comparison to young (Y) mice (Appendix A, *n* = 5–6). Average fold change is implicated by color scale. For individual LM, detectable only in one group, fold-change was calculated based on the corresponding LOD (Appendix A). Statistics: Data are shown as (**B**) mean ± SEM, (**D**–**G**) mean and (**C**) median (min to max); *p*-values were calculated by one-way ANOVA for multiple comparisons with Tukey’s posthoc test or Brown–Forsythe and Welch ANOVA with Dunnett’s T3 posthoc test (Appendix A). Statistical significance is indicated by asterisks for comparisons between CR and O, by hashes for comparisons between CR+RF and O and by ǂ for comparisons with the levels of young mice. * *p* ≤ 0.05, **/ǂǂ *p* ≤ 0.01, ǂǂǂ *p* ≤ 0.001, ns = not significant.

## Data Availability

All study data associated and materials are included in the article and the Appendix A.

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
