# Peer review of "Short-Term Caloric Restriction and Subsequent Re-Feeding Compromise Liver Health and Associated Lipid Mediator Signaling in Aged Mice"

_nutrients, 2023, doi:10.3390/nu15163660_

Round 1

Reviewer 1 Report

Introduction

1.     Intro needs a lot of work! The English is not great, it is jumpy (thoughts on the page) and hard to follow.

2.     Why do you need to use an abbreviation for lipid mediators? Using LM is hard for the reader as it is an uncommon abbreviation to use, and I think unnecessary.

Results

1.     In the second paragraph, the authors state that “Furthermore, typical proteomic markers for macrophage activation (CD14, CD36, CD68 and YM1) were amongst the most strongly and significantly upregulated proteins within the hepatic proteome of aged mice (Fig. 1C)”. However, looking at the graph, I do not agree with this statement, particularly that they are “the most strongly upregulated” ones.

2.     With the exception of maybe Ym1, there does not seem to be very good correlation between the volcano plot and the relative protein abundance shown in Figure D. In fact, CD68 is actually decreased in the relative abundance - how do you explain this? also what are the stars (****) that are on figure 1D?

3.     When reporting the results about resident macrophages at the end of page 5, the authors report that there is a decrease in F4/80, a marker of resident macrophages in the aged mice, yet in the next sentence discuss that there are alterations in the M2-specific markers of resident macrophages in the aged. How can there be no macs in the aged mice yet a a change in their polarisation?

4.     When discussing Figure 2B the authors say that there are changes (e.g. “…the total amount of LM derived from docosahexaenoic acid (DHA) dropped by about 40% as consequence of aging…” although there are no significant differences seen. This is the same for all of panel B and needs to be addressed.

5.     Again, there is a statement that “We found a substantial decrease of monohydroxylated LM due to aging” there are no stats on this figure to justify this statement.

6.     Liver ALT, AST and ALP are not measures of liver damage, only plasma values are. When liver cells are damaged they release these into the circulation and thus high levels are indicative of liver damage. They are part of normal metabolism within the liver and an increased level maybe associated with an increase in the capacity of that enzyme under that condition. Thus, the sentence on the top of page 7 (lines 272-274) is not a correct statement. This is also true when discussing the effects of CR on these parameters too (page 6, lines 345-9).

7.     With respect to the negative data presented in figure 1A and B, with old animals not showing any difference in inflammation when compared to control mice, why did the researchers continue with the CR component? It is not inflammaging as the title suggests.

8.     I don’t understand why the authors do not show the BW regain after the 2d refeeding as it is important but also a fair bit of weight to regain in 2 days! This should be included.

9.     In section 3.4 the authors again say that there is a decrease in metabolites between old and CR animals when there is no significant difference!

10.  Figure legends need to have how you indicate significance, i.e. stars

Discussion

1.     You start the second paragraph with they…. They who? What are you talking about?

2.     There are sections of the discussion that will need to be re-written with comments that I have stated above and thus have not gone through it.

The English throughout the manuscript needs work (in particular sentence and paragraph structure in the introduction). There is a lot of thoughts put down in a paragraph that dont flow well or link.

Please don't start new paragraphs with they! the reader has no idea what you are talking about

Author Response

see PDF attached.

Reviewer 2 Report

In this manuscript the authors presented interesting data that questions the benefits of short-term calorie restriction (CR) in the normal, physiologically aged mice. The authors did not find strong evidence for age-associated inflammation in the aged female mice (18 months old) compared to young (2-3 months old mice). Proteomic analysis showed some differences in the enzymes involved in the conversion of PUFAs to monohydroxylated derivatives (Fig. 1C). However, there were no differences in the LMs, which are further biosynthesized from the monohydroxylated precursors, even though the LM precursors were also decreased in the liver of aged mice (Fig 2C). CR increased levels of LM precursors, while CR+RF decreased LM precursors (Fig. 4D). Overall, while the data were mostly negative, the manuscript was well written, and the data raised interesting points regarding CR in a normal aged individual. There are some concerns regarding the discussion and presentation of data, listed below.

Major comments:

- In the authors’ aging cohorts, there were really no evidence for “inflammaging” in the liver. Thus, the title is a little misleading. The way the data was interpreted and discussed was rather focused in the context of inflammaging as well; however, the data could suggest that these aged female mice were still metabolically fit and resilient. Are there papers showing the effects of short-term CR on liver LMs in young mice?

- On that note, the differences seen in the proteomes and LMs in CR vs. CR+RF groups could be due to body weight difference? The authors mentioned that after 2 days of refeeding, the mice regained their body weight. Data were not shown but it would be useful to show that in Fig. 3A, to help the readers understand the implications of the data.

- What is the implication of reduced levels of LM precursors? The authors explained it somewhat, but it could be clarified further and discussed in the context of metabolic adaptation, given that this is the most striking difference between old and young (Fig. 1C), and old, CR and CR+RF (Fig 4D).

- The interpretation of CD163 data is confusing. This was grouped under M2 but also is a marker for Kupffer cells. The statements in lines 241-246 were contradictory.  

Author Response

see PDF attached.

Round 2

Reviewer 1 Report

Much improved

Much improved